# The TFIIH complex is required to establish and maintain mitotic chromosome structure

**Julian Haase[1], Richard Chen[1†‡], Wesley M Parker[1†], Mary Kate Bonner[1], Lisa M Jenkins[2], Alexander E Kelly[1*]**

[1]Laboratory of Biochemistry & Molecular Biology, National Cancer Institute, NIH, Bethesda, United States; [2]Laboratory of Cell Biology, National Cancer Institute, NIH, Bethesda, United States

**Abstract** Condensins compact chromosomes to promote their equal segregation during mitosis, but the mechanism of condensin engagement with and action on chromatin is incompletely understood. Here, we show that the general transcription factor TFIIH complex is continuously required to establish and maintain a compacted chromosome structure in transcriptionally silent *Xenopus* egg extracts. Inhibiting the DNA-dependent ATPase activity of the TFIIH complex subunit XPB rapidly and reversibly induces a complete loss of chromosome structure and prevents the enrichment of condensins I and II, but not topoisomerase II, on chromatin. In addition, inhibiting TFIIH prevents condensation of both mouse and *Xenopus* nuclei in *Xenopus* egg extracts, which suggests an evolutionarily conserved mechanism of TFIIH action. Reducing nucleosome density through partial histone depletion restores chromosome structure and condensin enrichment in the absence of TFIIH activity. We propose that the TFIIH complex promotes mitotic chromosome condensation by dynamically altering the chromatin environment to facilitate condensin loading and condensin-dependent loop extrusion.

**\*For correspondence:**
alexander.kelly@nih.gov

†These authors contributed equally to this work

**Present address:** ‡Department of Pharmacology, University of Texas, Southwestern Medical Center, Dallas, United States

**Competing interest:** The authors declare that no competing interests exist.

## Editor's evaluation

This paper reports the surprising observation that the general transcription factor TFIIH, but not transcription, is required for chromosome condensation in frog egg extracts. TFIIH may act by facilitating condensin localization and function. This opens up a lot of interesting new questions and lines of research that promise to add significantly to the field of chromosome biology. It will now be interesting to directly test the proposed mechanism of action, and to examine whether this role of TFIIH extends to somatic cells and other animals.

## Introduction

Entry into mitosis triggers dramatic structural changes in chromosomes that are required for their equal segregation (*Zhou and Heald, 2020*). A process known as chromosome condensation compacts sister chromatids into rod-like structures to promote individualization and prevent DNA damage during division. Sister chromatids form linear rod-like structures and separate from each other during prophase, and then shorten and thicken during prometaphase, ultimately forming fully condensed metaphase chromosomes. Electron microscope analyses of isolated mitotic chromosomes suggested that DNA is organized into loops around a protein scaffold that is concentrated along the central axis of each chromatid (*Earnshaw and Laemmli, 1983*; *Paulson and Laemmli, 1977*; *Paulson et al., 2021*). In agreement with this, recent studies using chromosome conformation capture and modeling

indicate that a sequential helical arrangement of loops is formed around a central axis (*Naumova et al., 2013*; *Gibcus et al., 2018*; *Elbatsh, 2019*), although alternative models of loop arrangement have been proposed (*Chu et al., 2020*). How these elements of chromosome structure are established and maintained during chromosome segregation remains a major question in biology.

The evolutionary conserved five-subunit condensin complex is an ATPase that serves as the primary driver of mitotic chromosome condensation (*Kinoshita and Hirano, 2017*). Many eukaryotes express two different forms of the condensin complex, condensin I and II (*Hirano et al., 1997*; *Ono et al., 2003*). Condensin I and II share two structural maintenance of chromosomes (SMC) subunits (SMC2 and SMC4) that form a V-shaped dimer with two ATP-binding head domains, but have distinct regulatory subunits (CAP-H/H2, CAP-D2/D3, and CAP-G/G2). Condensin I is only loaded onto chromosomes during mitosis, whereas condensin II is present on chromatin throughout the cell cycle (*Hirota et al., 2004*; *Ono et al., 2004*; *Walther et al., 2018*). During mitosis, both condensin complexes are concentrated along the central axis of individual chromatids, but have been demonstrated to have distinct roles in maintaining the lateral and axial compaction of chromatids (*Green et al., 2012*; *Samejima et al., 2012*; *Bakhrebah et al., 2015*). A recent study in chicken DT40 cells proposed that condensin II initially forms large DNA loops, which are then subdivided into smaller loops by condensin I (*Gibcus et al., 2018*). However, in *Xenopus* egg extracts, condensin II is not essential for the formation of condensed chromatids but remains important for optimal chromosome morphology and proper centromere function (*Shintomi and Hirano, 2011*; *Bernad et al., 2011*). Topoisomerase IIα (topo II) is an ATP-dependent DNA strand-passing enzyme that is required to individualize and resolve chromatids that is also involved in chromosome condensation (*Kinoshita and Hirano, 2017*). Like the condensins, topo II is localized to the mitotic chromatid axis, and current models suggest it acts to control the axial compaction of chromosomes through the regulation of condensin-mediated loops (*Nielsen et al., 2020*; *Shintomi and Hirano, 2021*; *Paulson et al., 2021*).

Multiple factors promote the association of condensin with chromatin (*Robellet et al., 2017*), including but not limited to, histones (*Ball et al., 2002*; *Liu et al., 2010*; *Tada et al., 2011*), nucleosome-free regions (*Piazza et al., 2014*; *Toselli-Mollereau et al., 2016*), single-stranded DNA (*Sakai et al., 2003*; *Sutani et al., 2015*), and supercoiling (*Kimura et al., 1998*; *Kim et al., 2021*). Tuning condensin levels on chromatin is important for proper chromosome condensation, as prior studies have shown that altering condensin levels leads to gross changes in the axial and lateral compaction of chromosomes, as well as the alteration of the size of extruded DNA loops (*Shintomi and Hirano, 2011*; *Thadani et al., 2018*; *Elbatsh, 2019*; *Choppakatla et al., 2021*; *Goloborodko et al., 2016a*; *Fitz-James et al., 2020*; *Goloborodko et al., 2016b*). However, how condensin levels and activity on chromosomes are regulated remains an outstanding question. One conserved feature of condensins across multiple eukaryotes is their enrichment at sites of RNA polymerase II (Pol II)-dependent transcription (*Bernard and Vanoosthuyse, 2015*; *D'Ambrosio et al., 2008*; *Schmidt et al., 2009*; *Nakazawa et al., 2015*; *Sutani et al., 2015*; *Kim et al., 2016*; *Kranz et al., 2013*; *Kim et al., 2013*; *Dowen et al., 2013*). This suggests a potentially conserved role for transcription or the transcriptional machinery in regulating condensin loading and function during mitosis. However, the role of transcription in mitotic chromosome condensation is not well understood (*Robellet et al., 2017*). Here, we show that the TFIIH complex, part of the transcriptional pre-initiation complex, is continuously required for chromosome condensation by promoting condensin loading and function on chromatin.

## Results

### The TFIIH complex is required for chromosome condensation in *Xenopus* egg extracts

To examine possible relationships between transcription-associated activities and mitotic chromosome condensation, we used cell-free *Xenopus* M phase egg extracts, a system in which condensin function is well characterized and in which zygotic transcription is inactive (*Kinoshita et al., 2015*; *Hirano and Mitchison, 1994*; *Barrows and Long, 2019*; *Chen et al., 2019*; *Amodeo et al., 2015*; *Jukam et al., 2017*). As previously demonstrated (*Hirano and Mitchison, 1993*; *Shintomi et al., 2015*), sperm nuclei incubated with a high-speed supernatant (HSS) of egg extracts first swell during the exchange of protamines for maternal histones, assemble into a diffuse 'cloud' of chromatin, and gradually form clusters of condensed single-chromatid structures (*Figure 1A*). As expected, depletion

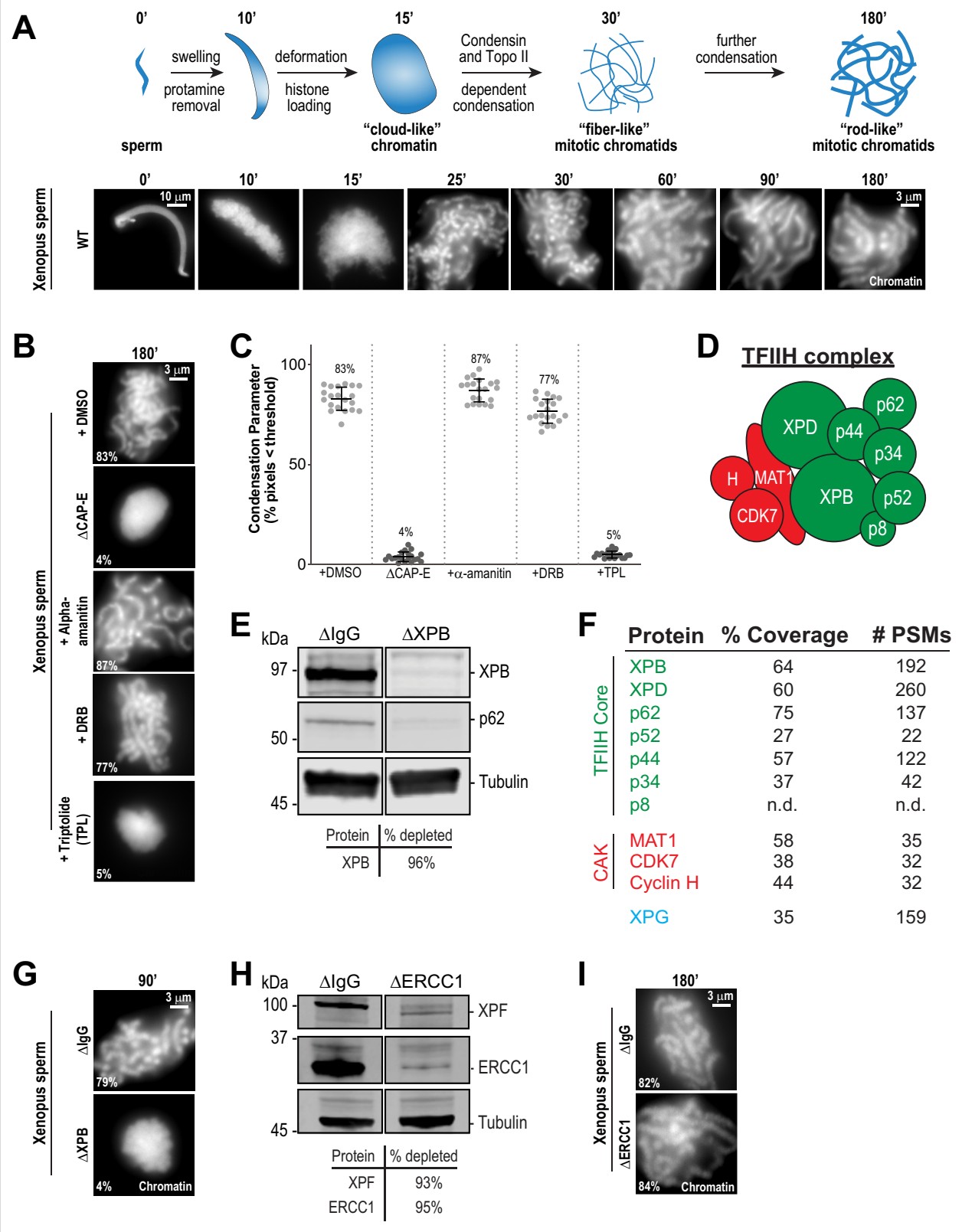

**Figure 1.** The TFIIH complex is required for chromosome condensation in Xenopus egg extracts. (**A**) Top: Schematic of chromatid assembly in M-phase high-speed supernatant (HSS) extract. Bottom: Representative fluorescence images of fixed samples from a chromatid assembly reaction taken at indicated timepoints after sperm addition. Chromatin was stained with Hoechst. (**B**) Representative fluorescence images of chromatid assembly at steady state (180 min after sperm addition) in the presence of indicated inhibitors. Each drug or DMSO control was added at 25 min to exclude effects

*Figure 1 continued on next page*

*Figure 1 continued*

on the protamine-histone exchange process. Final inhibitor concentrations: Triptolide (50 µM), DRB (100 µM), and a-amanitin (54 µM). See **Figure 1—figure supplement 1A** for CAP-E depletion. Mean condensation parameters for each condition are indicated in lower left corner of each image. (**C**) Scatter plot of the percentage of pixels below a threshold of 35% of image maximum fluorescence intensity (the condensation parameter), which measures the progressive change in the fluorescence intensity distribution that occurs during condensation, for each condition. Error bars represent SD, and the mean values are indicated. n = 20 structures for each condition. Two biological replicates were performed, quantified structures are from a single experiment. (**D**) Schematic of the TFIIH complex. TFIIH core complex is green, CAK subcomplex in red. (**E**) Western blot for XPB, p62, and Tubulin in IgG or XPB-depleted extracts. (**F**) TFIIH complex members that interact with XPB in M-phase HSS extract, as identified by mass spectrometry. Purifications were performed in the absence of chromatin. PSMs = peptide spectrum matches. (**G**) Representative fluorescence images of chromatid assembly 90 min after sperm addition in IgG or XPB depleted extracts. Chromatin was stained with Hoechst. (**H**) Western blot for XPF, ERCC1, and Tubulin in IgG or ERCC1-depleted extracts. (**I**) Representative fluorescence images of chromatid assembly 180 min after sperm addition in IgG or ERCC1-depleted extracts. Chromatin was stained with Hoechst.

The online version of this article includes the following source data and figure supplement(s) for figure 1:

**Source data 1.** Source data for **Figure 1E and H**.

**Figure supplement 1.** Immunodepletions and the effects of RNases and various transcription inhibitors on chromosome condensation and cell cycle state.

**Figure supplement 1—source data 1.** Source data for **Figure 1—figure supplement 1A, B,D, E, G**.

**Figure supplement 2.** Effects of various transcription inhibitors on RNA Polymerase II C-terminal phosphorylation at centromeres.

**Figure supplement 3.** Condensation parameters.

of condensins I and II with anti-CAP-E antibodies caused a severe loss of condensation (**Figure 1B** and **Figure 1—figure supplement 1A**), and this effect could be quantified using a previously described 'condensation parameter' (**Figure 1C**; **Maddox et al., 2006**). We then asked if various inhibitors of the Pol II-dependent transcriptional machinery affect chromosome condensation in egg extracts when added at the beginning stages of the condensation process (t = 25'), after protamine exchange and 'cloud' formation (**Figure 1A**; **Shintomi et al., 2015**). Inhibitors of RNA polymerase II elongation, such as α-amanitin and 5,6-dichloro-1-beta-D-ribofuranosylbenzimidazole (DRB) had no effect on chromosome condensation (**Figure 1B and C**), but induced the loss of the actively elongating (S2ph) form of RNA polymerase II at centromeres as expected (**Figure 1—figure supplement 2A and B**; **Grenfell et al., 2016**; **Blower, 2016**; **Perea-Resa et al., 2020**). In addition, condensation was not altered by pretreatment of extracts with RNase A to remove endogenous RNAs (**Figure 1—figure supplement 1B and C** and **Figure 1—figure supplement 3A**) or by pretreatment with RNase H to remove RNA-DNA hybrids (**Figure 1—figure supplement 1C** and **Figure 1—figure supplement 3A**; **Thakur and Henikoff, 2020**). These findings, combined with the fact that very little transcription occurs in *Xenopus* embryos before the maternal-to-zygotic transition (**Jukam et al., 2017**; **Chen et al., 2019**), indicates that active transcription is not required for chromosome condensation.

In contrast, treatment of extracts with the transcription initiation inhibitor triptolide completely blocked chromosome condensation, resulting in decondensed chromatin masses similar to structures observed upon condensin depletion (**Figure 1B and C**; **Ono et al., 2003**; **Shintomi and Hirano, 2011**). Triptolide is an inhibitor of the multi-subunit TFIIH complex (**Figure 1D**), which as part of the pre-initiation complex is required for Pol II-dependent transcription initiation (**Titov et al., 2011**; **Compe and Egly, 2012**). Triptolide covalently binds to the XPB subunit of TFIIH and blocks its DNA-dependent ATPase (**Titov et al., 2011**). The ATPase activity of XPB is required for the DNA translocase function of TFIIH, which in collaboration with the pre-initiation complex, is essential to twist and open DNA to allow RNA polymerase II to productively engage the template strand to initiate transcription (**Fishburn et al., 2015**; **Tomko et al., 2017**; **Kim et al., 2000**; **Grünberg et al., 2012**; **Nogales and Greber, 2019**; **Schilbach et al., 2021**). In agreement with these described functions of TFIIH, triptolide prevented the accumulation of both paused (S5ph) and actively elongating (S2ph) forms of Pol II at centromeres (**Figure 1—figure supplement 2A and B**).

To rule out off-target effects of triptolide, we also examined chromosome condensation in extracts depleted of XPB ( > 95%) using anti-XPB antibodies (**Figure 1E** and **Figure 1—figure supplement 1D**). XPB depletion also depleted the TFIIH subunit p62 (**Figure 1E**), and mass spectrometry analysis of proteins co-depleted with XPB identified strong enrichment of other core TFIIH complex members (**Figure 1F**). As shown in **Figure 1G**, XPB depletion completely abrogated chromosome condensation

(*Figure 1—figure supplement 3B*), thus confirming that XPB is the physiological target of triptolide and demonstrating an unprecedented role for the TFIIH complex in chromosome condensation.

The TFIIH complex is involved in processes other than transcription initiation that could also potentially affect chromosome condensation, including the nucleotide excision repair (NER) pathway (*Compe and Egly, 2012*). Consistent with a previous report (*Ito et al., 2007*), we identified the NER endonuclease XPG as a strong interactor with the TFIIH complex (*Figure 1F*). During NER, XPG and ERCC1-XPF, another NER endonuclease, act together to resect DNA lesions after the XPB and XPD subunits of TFIIH have unwound DNA around the lesion (*Kolesnikova et al., 2019*; *Compe and Egly, 2012*). To address whether the NER pathway is required for condensation, we immunodepleted ERCC1 from egg extracts, which prevents NER (*Klein Douwel et al., 2017*). Depletion of the ERCC1/XPF complex did not perturb chromosome condensation (*Figure 1H and I*, *Figure 1—figure supplement 1E*, and *Figure 1—figure supplement 3C*), indicating that the TFIIH complex does not act through the NER pathway to promote chromosome condensation.

TFIIH also associates with the three protein CAK complex (CDK-Activating Kinase) (*Figure 1F*), which contains the kinase CDK7. CDK7 directly phosphorylates the Pol II C-terminus at Ser5 (Pol II S5ph), and is required for the release of Pol II from the pre-initiation complex to promote transcriptional pausing and other transcriptional events (*Rimel and Taatjes, 2018*; *Glover-Cutter et al., 2009*; *Akhtar et al., 2009*; *Ebmeier et al., 2017*). As expected, inhibition of CDK7 with THZ1 (*Kwiatkowski et al., 2014*) induced the loss of the paused and elongating forms of Pol II from centromeres (*Figure 1—figure supplement 2A* and B). However, THZ1 treatment had no effect on condensation (*Figure 1—figure supplement 1F* and *Figure 1—figure supplement 3D*). Furthermore, inhibition of CDK7 did not rescue the condensation defects observed in triptolide-treated extracts, ruling out abnormal upregulation of CDK7 activity upon inhibition of XPB (*Figure 1—figure supplement 1F* and *Figure 1—figure supplement 3D*). CDK7, as part of the labile CAK complex, is also required for the initial activation of CDK1 to promote entry into mitosis (*Larochelle et al., 2007*), and thus we tested the possibility that triptolide exerts its effects on condensation through down-regulation of CDK1 activity. However, triptolide did not affect CDK1 activity or M phase maintenance in HSS extracts, as indicated by unaltered CDK1 activation loop phosphorylation (T161ph) and histone H3T3ph levels (*Kelly et al., 2010*; *Figure 1—figure supplement 1G*). Thus, the ATPase activity of the TFIIH complex is required for condensation in a manner that is independent of TFIIH activities in NER, active transcription, and the cell cycle.

## The TFIIH complex is required for the dynamic maintenance of mitotic chromosome structure

Chromosome condensation must be actively maintained during mitosis (*Kinoshita et al., 2015*), but prior work has suggested that condensation establishment and maintenance may involve distinct mechanisms (*Nielsen et al., 2020*; *Hirano and Mitchison, 1993*). To ask if the TFIIH complex is required for the maintenance of chromosome condensation as well as its establishment, we added triptolide at various timepoints during the condensation process: at 25 min after addition of sperm to extracts, at condensation initiation; at 60 min, when chromosomes were partially condensed; and at 180 min, when chromosomes were in a steady state of full condensation (*Figure 2A*). In DMSO-treated controls, condensation increased steadily during these times until reaching a plateau at 180 min (*Figure 2A*, and *Figure 2—figure supplement 1A*). However, addition of triptolide at any of these times during the condensation process, including at 180 min, induced the rapid (less than 5 min) loss of discernable individual chromatids and a diffuse DNA staining pattern, similar to the structures observed upon condensin depletion (*Figure 1B*), that persisted throughout the rest of the timecourse (*Figure 2A*, and *Figure 2—figure supplement 1A*). Thus, TFIIH activity is required for maintaining chromosome condensation as well as its establishment.

To test if the triptolide-induced loss of mitotic chromosome structure is reversible, we allowed chromosomes to completely condense for 180 min, treated with triptolide for 30 min to induce loss of chromosome structure, and then diluted 10-fold with extracts containing triptolide or DMSO (*Figure 2B*). Dilution with DMSO-treated extracts resulted in the rapid regain (within 5 min) of chromatid structure, while chromosomes remained uncondensed upon dilution with triptolide-treated extract (*Figure 2B and C*, *Figure 2—figure supplement 2A and B*). These results demonstrate that the effects of TFIIH inhibition on chromosome condensation can be reversed rapidly

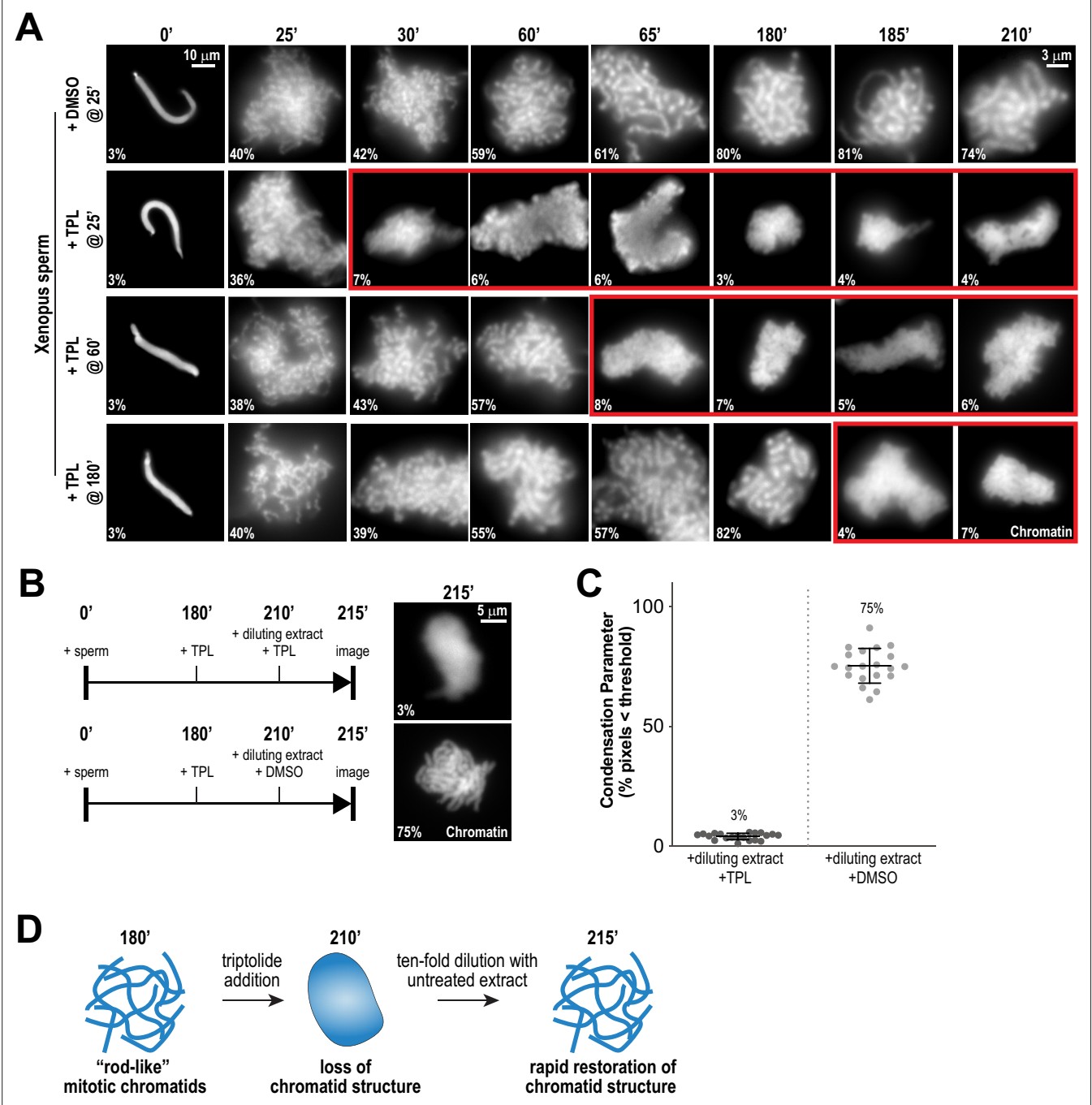

**Figure 2.** Continuous TFIIH activity is required to maintain chromosome structure. (**A**) Representative fluorescence images of chromatid assembly reactions sampled at indicated timepoints. DMSO or triptolide (TPL) were added at indicated times, immediately after sample retrieval for imaging. Chromatin was stained with Hoechst. Red boxes indicate representative images of decondensed chromosomes after triptolide addition. Mean condensation parameters for each condition are indicated in lower left corner of each image, see **Figure 2—figure supplement 1A** for more details. (**B**) Representative fluorescence images of chromatid masses fixed and spun onto coverslips 215 min after sperm addition, subjected to the indicated manipulations. Nuclei were incubated in extracts for 180 min to allow for the formation of fully condensed chromatids, followed by addition of DMSO or triptolide. After 30 min of incubation with DMSO/triptolide, reactions were diluted 10-fold with extract containing DMSO or triptolide, and further incubated for 5 min before taking samples for imaging. See **Figure 2—figure supplement 2** for additional controls. Chromatin was stained with Hoechst. (**C**) Scatter plots of the percentage of pixels below a threshold of 35% of image maximum fluorescence intensity (the condensation parameter), for each condition in (**B**). Error bars represent SD, and the mean values are indicated. n = 20 structures for each condition. Two biological replicates were performed, quantified structures are from a single experiment. (**D**) Schematic illustrating the effects of triptolide addition and dilution on chromosome structure maintenance and reformation.

*Figure 2 continued on next page*

*Figure 2 continued*

The online version of this article includes the following figure supplement(s) for figure 2:

**Figure supplement 1.** Condensation parameters of condensation kinetics after DMSO or Triptolide treatment.

**Figure supplement 2.** Continuous TFIIH activity is required to maintain chromosome structure and condensin levels on chromatin.

(*Figure 2D*), suggesting that the TFIIH complex acts continually in dynamic processes required for condensation.

## TFIIH is required for the enrichment of condensins on mitotic chromatin

The proper assembly of mitotic chromosomes in egg extracts requires the activities of condensins I and II and Topoisomerase IIα (topo II), which localize along condensed chromatids on a central axis (*Samejima et al., 2012*; *Ono et al., 2003*; *Hirano and Mitchison, 1993*; *Shintomi et al., 2015*; *Earnshaw and Heck, 1985*). To test the hypothesis that TFIIH is involved in this localization, we used immunofluorescence to measure levels of XPB, condensin I/II, and topo II on chromatin in the presence or absence of triptolide (added at t = 25′, imaged at t = 180′). In control extracts, the XPB was present in a diffuse pattern over the chromatid mass (*Figure 3A*), with discrete puncta of XPB visible along the entire length of isolated chromatids (*Figure 3A* (inset)). Condensins I and II and topo II were each present on the central axis of each chromatid, as previously reported (*Ono et al., 2003*; *Maeshima and Laemmli, 2003*; *Figure 3A*). Triptolide treatment severely reduced XPB levels on chromatid clusters (*Figure 3B*; 21% of control), indicating that inhibition of the XPB ATPase affects its association with chromatin. Strikingly, triptolide also caused significant reductions in the levels of CAP-E (37% of control) and CAP-C (31% of control), which are subunits of both condensin I and II, and also reduced levels of the condensin II-specific subunit CAP-D3 (*Figure 3B*; 25% of control). The condensin that remained on chromosomes was present in a diffuse pattern with no defined axis (*Figure 3A*). Since the amount of condensin I in extracts is three- to fivefold higher than that of condensin II, and since condensin I is the major driver of chromosome condensation in egg extracts (*Ono et al., 2003*; *Shintomi and Hirano, 2011*), we conclude that triptolide prevents the proper loading and enrichment of both condensin I and condensin II. Conversely, triptolide had no significant effect on topo II levels (*Figure 3A and B*; 93% of control), although topo II was no longer concentrated on a central axis. A similar effect on topo II localization is seen upon condensin depletion from egg extracts (*Shintomi et al., 2017*). Similarly, SUPT16H, a member of the FACT complex implicated in both transcription (*Zhou et al., 2020*) and chromosome condensation (*Shintomi et al., 2015*), was present at similar levels in the presence or absence of triptolide (*Figure 3A and B*; 113% of control). CAP-C levels also decreased significantly (61% of control) when fully condensed chromosomes were treated with triptolide at t = 180′ for 35 min (*Figure 2—figure supplement 2C*). These results demonstrate that the TFIIH complex is required for the enrichment of condensin I/II on chromosomes during condensation establishment and maintenance. They also suggest that the TFIIH complex specifically regulates condensins and not topo II, since TFIIH inhibition alters condensin but not topo II levels on chromatin (*Figure 3B*) and causes defects in condensation that are similar to condensin depletion but not topo II inhibition (*Ono et al., 2003*; *Shintomi et al., 2017*; *Hirano and Mitchison, 1993*).

In reciprocal experiments, we found that XPB localization to chromatin was maintained following the depletion of condensins (*Figure 3—figure supplement 1A*). Since both TFIIH and condensin levels on chromatin decreased in response to triptolide (*Figure 3B*), we hypothesized that TFIIH might promote recruitment of condensin complexes to chromosomes through physical interaction. We found that the condensin II subunit CAP-D3 co-immunoprecipitated with XPB from high-speed supernatants of egg extracts (*Figure 3—figure supplement 1B*). However, XPB was not co-immunoprecipitated with anti-CAP-E antibodies, even though we observed robust interactions with other condensin I and II subunits including CAP-D3 (*Figure 3—figure supplement 1C*). These results suggest that any interactions between TFIIH and condensins are weak, and not necessarily direct. Furthermore, TFIIH only partially co-localized with condensins on single chromatids, and the TFIIH complex did not demonstrate the stereotypical axial staining pattern observed for condensins, which argues against a scaffolding role for TFIIH (*Figure 3—figure supplement 1D*). Altogether, these results show that the TFIIH complex acts upstream of condensin enrichment and weakly associates with the CAP-D3 subunit of condensin II, but indicate that the main role of TFIIH in condensin enrichment is not through direct recruitment.

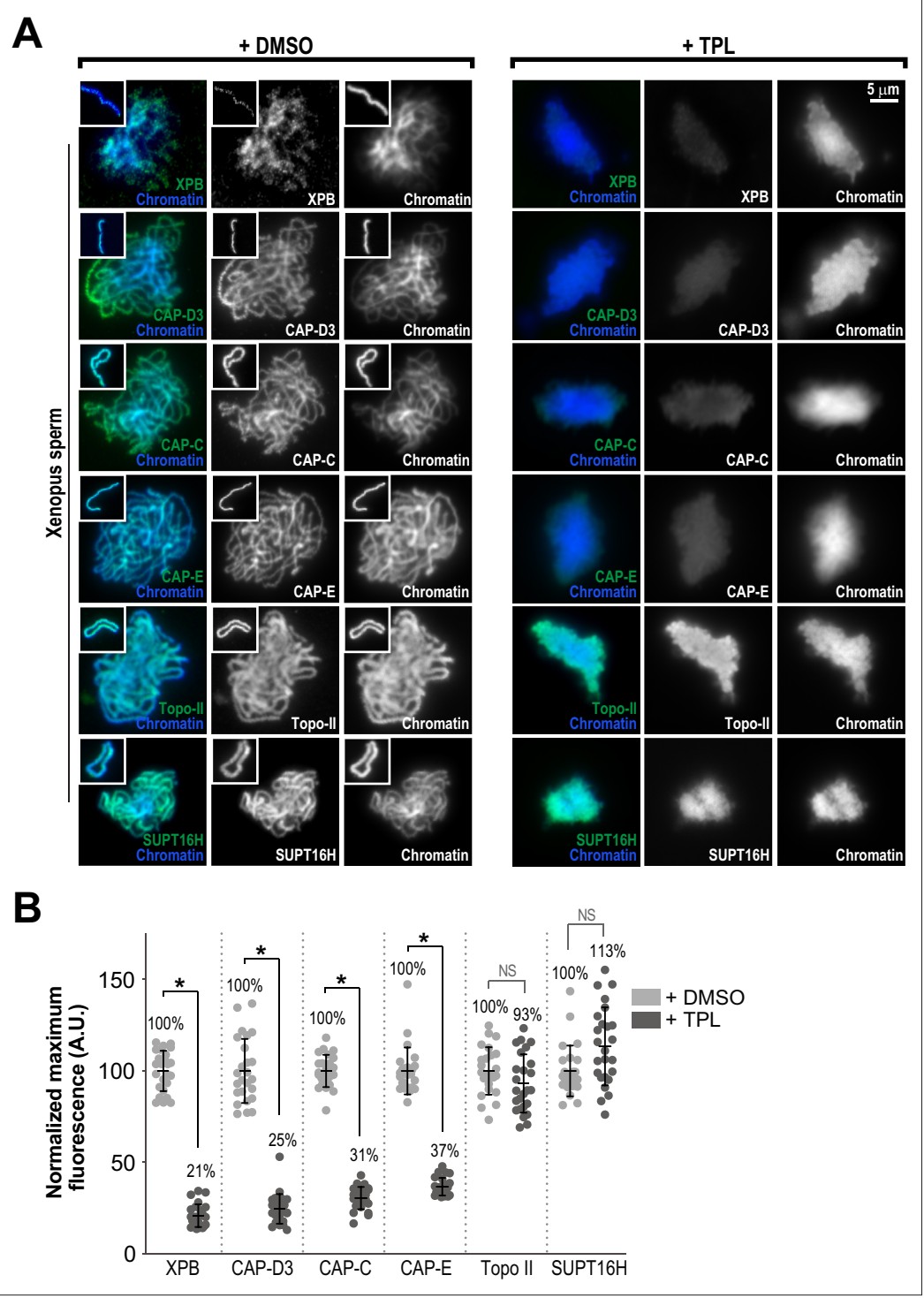

**Figure 3.** The TFIIH complex is required for the enrichment of condensins on chromosomes. (**A**) Representative immunofluorescence images of *Xenopus* sperm nuclei incubated with DMSO or triptolide (TPL) treated extracts for 180 min. Chromatids were labeled with Hoechst and anti-XPB, anti-CAP-D3 (condensin II), anti-CAP-C (condensin I & II), anti-CAP-E (condensin I and II), anti-SUPT16H (FACT complex), or anti-Topo II antibodies. Images of individual chromatids are shown in insets. (**B**) Quantification of fluorescence intensity of indicated proteins from experiment in (**A**), normalized to DMSO. n = 50 structures for each condition. Error bars represent SD, and asterisks indicate a statistically significant difference (*, p < 0.001), NS indicates no statistical significance. A.U., arbitrary units. Two biological replicates were performed, quantified structures are from a single experiment.

*Figure 3 continued on next page*

*Figure 3 continued*

The online version of this article includes the following source data and figure supplement(s) for figure 3:

**Figure supplement 1.** The TFIIH complex is required for the enrichment of condensins on chromosomes.

**Figure supplement 1—source data 1.** Source data for *Figure 3—figure supplement 1B and C*.

## TFIIH inhibition perturbs chromosome condensation before condensin and TFIIH are lost from chromosomes

To further investigate how TFIIH promotes condensation and the enrichment of condensins on chromatin, we determined the kinetics of condensin enrichment with and without TFIIH inhibition. Specifically, we added triptolide 25 min after addition of sperm nuclei to egg extracts and monitored chromatin morphology and condensin levels on chromatin over time (*Figure 4A*). As expected, the addition of triptolide led to rapid decondensation (*Figures 2A and 4B*, and *Figure 4—figure supplement 1A*), and an amorphous cloud of chromatin without discernable chromatids was seen within five minutes of drug addition. DMSO-treated extracts progressed in a stereotypical fashion, with increasing condensation over time (*Figure 4B* and *Figure 4—figure supplement 1A*). In contrast,

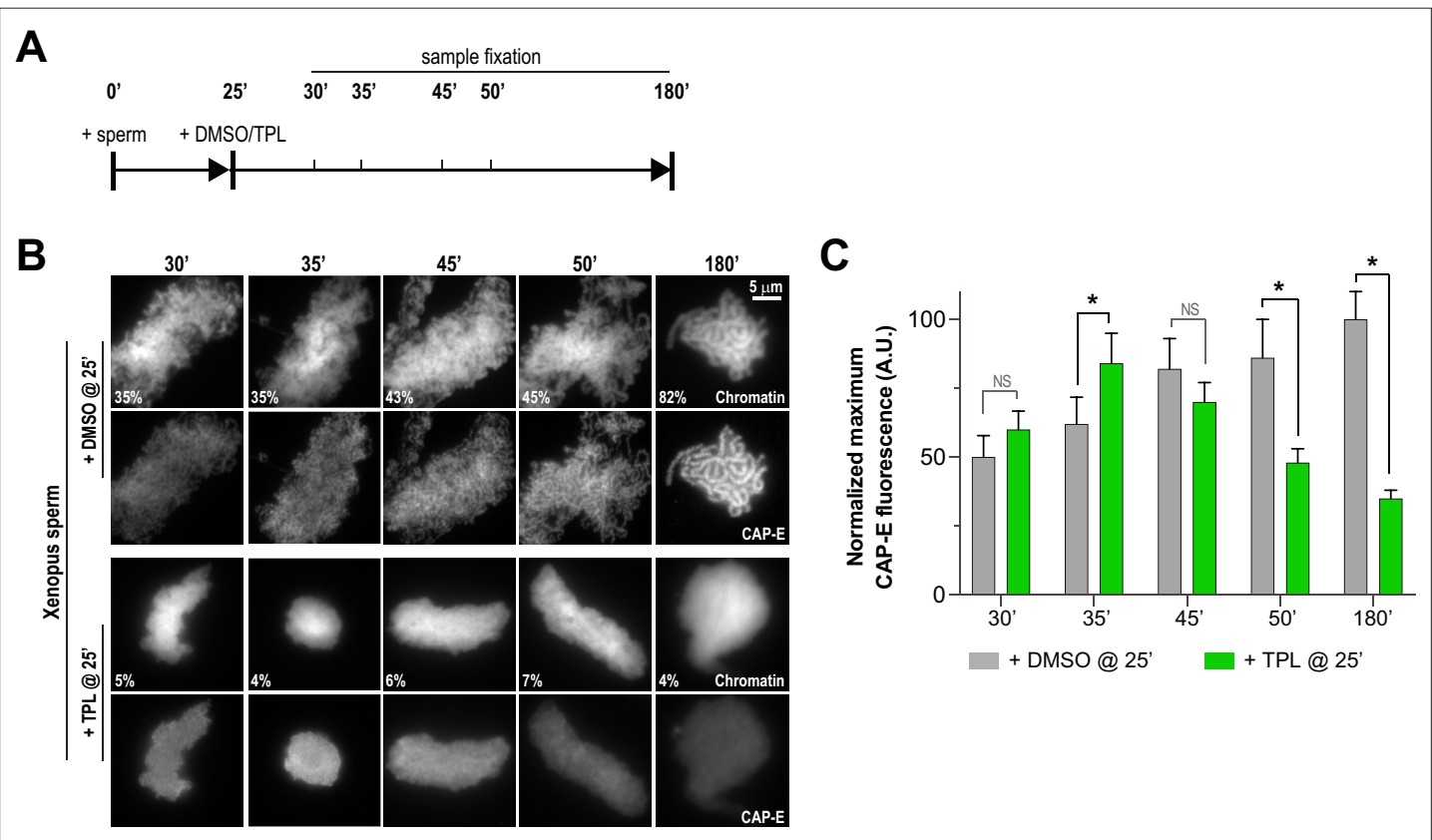

**Figure 4.** Triptolide perturbs condensation prior to its effects on condensin levels. (**A**) Schematic of assay to test the timing of triptolide-induced defects in condensation and condensin levels. After a twenty-five minute incubation, either DMSO or triptolide (TPL) was added, and samples were taken at indicated times and processed for immunofluorescence. (**B**) Representative immunofluorescence images of DMSO or triptolide treated extracts at indicated timepoints. Chromatids were labeled with Hoechst and anti-CAP-E (condensin I and II). Note that condensation is already lost in in the first timepoint after triptolide addition, as in *Figure 2A*. Mean condensation parameters for each condition are indicated in lower left corner of each image. (**C**) Quantification of fluorescence intensity of CAP-E from experiment in (**B**), normalized to the 180 min DMSO-treated sample. n = 50 structures for each condition. Error bars represent SD, and asterisks indicate a statistically significant difference (*, p < 0.001). A.U., arbitrary units. Two biological replicates were performed, quantified structures are from a single experiment.

The online version of this article includes the following figure supplement(s) for figure 4:

**Figure supplement 1.** Condensation parameters of condensation kinetics after DMSO or Triptolide treatment.

**Figure supplement 2.** Triptolide perturbs condensation prior to its effects on condensin levels.

total condensin levels, as monitored by CAP-E staining, increased for the first 10 min after triptolide addition, remaining similar to controls within the first 20 min before decreasing to 35% of control over the next 135 min (*Figure 4B and C*, and *Figure 4—figure supplement 2A*). Similarly, condensin II and XPB remained at control levels for 10 min after triptolide addition before decreasing relative to controls (*Figure 4—figure supplement 2B* and C). These results indicate that TFIIH plays an immediate role in promoting condensation, and is also required for the ongoing enrichment of condensins on chromosomes that occurs throughout the condensation process.

## Artificial formation of nucleosome-free regions bypasses the requirement for TFIIH in chromosome condensation

Several lines of evidence suggest that nucleosomes are a barrier to condensin loading and function on chromosomes. In *Xenopus* egg extracts, condensins display a strong preference for binding to DNA over nucleosomes (*Zierhut et al., 2014*; *Shintomi et al., 2017*; *Choppakatla et al., 2021*). The creation of nucleosome-free regions has been proposed to be important for chromosome condensation in fission yeast (*Toselli-Mollereau et al., 2016*), and double-stranded DNA has been shown to preferentially activate the ATPase activity of budding yeast condensin (*Piazza et al., 2014*). The TFIIH complex, which acts as a DNA translocase and facilitates changes to chromatin, might promote condensin binding and condensation by altering the chromatin environment to allow the exposure of DNA. To test this suggestion, we artificially introduced nucleosome-free regions on chromosomes by using anti-H4K12ac antibodies to partially deplete histones H3 and H4 from egg extracts (*Zierhut et al., 2014*) and examined the effects of triptolide on chromosome condensation. Since *Xenopus* sperm retain paternally derived histones H3 and H4 on chromatin (*Shintomi et al., 2015*), we used mouse sperm, which have previously been shown to condense in a condensin-dependent manner in *Xenopus* egg extracts (*Shintomi et al., 2017*) but which contain very low levels of histones H3 and H4 (*Brykczynska et al., 2010*). We found that triptolide inhibits the condensation of mouse sperm nuclei incubated in *Xenopus* egg extracts (*Figure 5A and B*, *Figure 5—figure supplement 1A*, and *Figure 5—figure supplement 2A*), indicating that TFIIH activity is required for condensation regardless of chromatin source. Control depletions with IgG did not perturb condensation, nor did it alter the inhibitory effects of triptolide (*Figure 5A and B*, *Figure 5—figure supplement 1B*, and *Figure 5—figure supplement 2A*). Remarkably, depletion of total histone H3 levels in extracts by only 24%, which reduced histone H3 levels on chromosomes by 38% (*Figure 5B and C*, and *Figure 5—figure supplement 1B*), rescued chromosome condensation in the presence of triptolide (*Figure 5A* and *Figure 5—figure supplement 2A*). Importantly, chromosomes were resistant to decondensation ten minutes after TFIIH inhibition, suggesting a direct effect of histone H3/H4 depletion (*Figure 5A*, and *Figure 5—figure supplement 2A*). Triptolide treatment caused substantial reductions in both XPB and condensin I/II subunit CAP-E on mouse chromosomes (respectively to 23% and 40% of control levels, *Figure 5B and C*), as it did with *Xenopus* chromosomes (*Figure 3A and B*). However, partial depletion of histones H3 and H4 resulted in a substantial increase in CAP-E levels (to 79% of control levels), and CAP-E labeled the chromatid axis as in controls (*Figure 5B and C*). In contrast, histone H3/H4 depletion did not suppress the triptolide-induced loss of XPB levels from chromosomes (*Figure 5B and C* and *Figure 5—figure supplement 1B*), indicating that histone depletion does not affect the inhibition of TFIIH by triptolide. The finding that TFIIH function in condensation can be bypassed by partial histone depletion strongly suggests that the main role of TFIIH in condensation is to alter the chromatin environment, possibly by regulating nucleosome composition and/or inhibitory proteins, which in turn promotes condensin function and enrichment on chromatin.

## Discussion
### Roles of the TFIIH complex in chromosome condensation

The prevailing model of chromosome condensation posits that condensins extrude loops of chromatin in an ATP-dependent manner to promote the formation of multiple intra-chromatid contacts (*Davidson and Peters, 2021*). Nucleosomes may form a barrier to both condensin loading and loop extrusion, since condensins bind more strongly to nucleosome-free DNA than to nucleosomes (*Zierhut et al., 2014*; *Shintomi et al., 2017*; *Toselli-Mollereau et al., 2016*; *Choppakatla et al., 2021*) and the binding to free DNA activates condensin ATPase activity (*Piazza et al., 2014*). In vitro

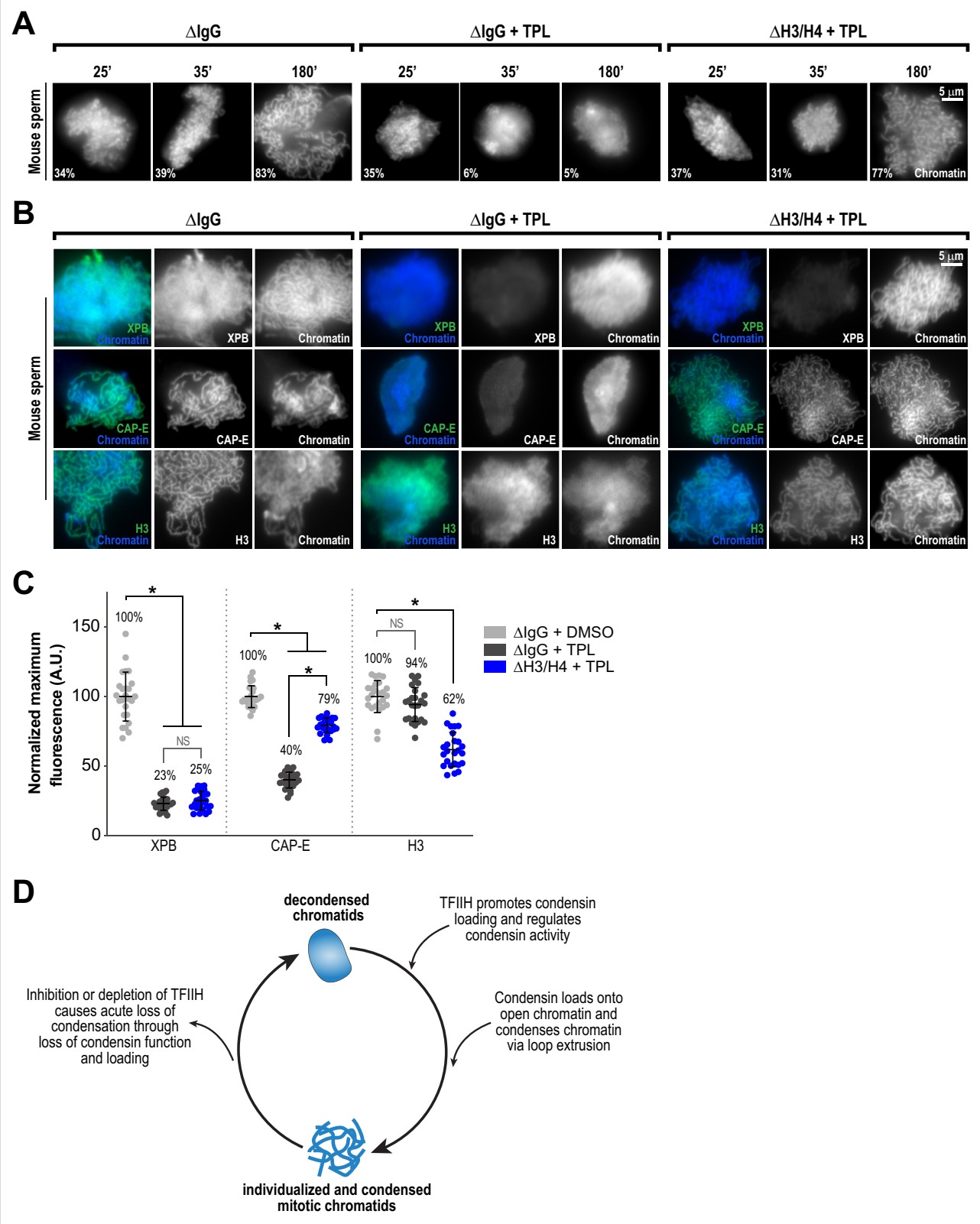

**Figure 5.** Partial histone H3/H4 depletion rescues chromosome condensation in the absence of TFIIH activity. (**A**) Representative fluorescence images of mouse sperm nuclei incubated with ΔIgG or ΔH4K12ac extracts and spun onto coverslips at indicated timepoints. DMSO or Triptolide were added at t = 25', immediately after sample acquisition for imaging. Chromatin was stained with Hoechst. Mean condensation parameters for each condition are indicated in lower left corner of each image. (**B**) Representative immunofluorescence images of mouse sperm nuclei incubated with ΔIgG or ΔH4K12ac

*Figure 5 continued on next page*

*Figure 5 continued*

extracts for 180 min. DMSO or triptolide were added at t = 25'. Chromatids were labeled with anti-XPB, anti-CAP-E (condensin I and II), or anti-histone H3, and co-stained with Hoechst. (**C**) Quantification of fluorescent intensity of indicated proteins from (**B**), normalized to DMSO. n = 50 structures for each condition. Error bars represent SD, and asterisks indicate a statistically significant difference (*, p < 0.001). A.U., arbitrary units. Two biological replicates were performed, quantified structures are from a single experiment. (**D**) Model depicting the role of the TFIIH complex in regulating condensin dynamics. Our data are consistent with a model in which TFIIH alters the chromatin environment to allow condensins to load and extrude loops of DNA.

The online version of this article includes the following source data and figure supplement(s) for figure 5:

**Figure supplement 1.** Effects of triptolide and histone depletion on condensation using mouse sperm.

**Figure supplement 1—source data 1.** Source data for *Figure 5—figure supplement 1B*.

**Figure supplement 2.** Condensation parameters of condensation kinetics after DMSO or Triptolide treatment.

experiments have reported that condensin can extrude loops of chromatinized DNA, but nucleosomes were present in these studies at low densities sufficient to expose naked DNA and permit condensin loading (*Kong et al., 2020*). Because mitotic chromatin in vivo is more compact and nucleosome dense (*Ou et al., 2017*; *Grigoryev et al., 2016*), it may need to be modified before condensin can load and extrude loops. Here, we have shown that mitotic chromosome condensation and condensin loading in *Xenopus* egg extracts require the TFIIH complex, and that TFIIH's roles in these processes can be bypassed by reducing nucleosome density. These findings provide further support for the idea that nucleosomes inhibit condensin function, and suggest that the TFIIH complex regulates the exposure of nucleosome-free DNA where condensin can load and function (*Figure 5D*). Chromosome condensation depends on condensin loading as well as condensin-mediated loop extrusion, and previous experiments have suggested that these events can be separated (*Goloborodko et al., 2016a*; *Kong et al., 2020*; *Kinoshita et al., 2015*; *Thadani et al., 2018*; *Ganji et al., 2018*). Our results suggest that TFIIH activity promotes both steps, since TFIIH inhibition rapidly stalls the condensation process before it impacts levels of condensins on chromatin (*Figure 4* and *Figure 4—figure supplement 2*), although it remains unclear whether TFIIH regulates these processes directly, or indirectly through removal of inhibitory proteins (see below). Since histone depletion restores both condensation and condensin loading in the absence of TFIIH activity (*Figure 5A–C*), it is likely that TFIIH-dependent changes to the chromatin environment are also required for condensin-mediated loop extrusion and chromatid formation. These changes could promote loop extrusion by creating additional binding sites necessary for intra-chromatid contacts and condensin processivity (*Goloborodko et al., 2016b*; *Goloborodko et al., 2016a*), or through regulation of condensin turnover (*Kinoshita et al., 2015*; *Thadani et al., 2018*). However, it is possible that TFIIH also promotes condensation in a condensin-independent manner.

During transcription initiation, the XPB subunit of TFIIH acts as a DNA translocase to help 'melt' promoter DNA to allow for Pol II to engage the template strand (*Nogales and Greber, 2019*; *Aibara et al., 2021*). Our studies with triptolide, which inhibits the ATPase activity of XPB required for translocation (*Titov et al., 2011*; *Kappenberger et al., 2020*; *Tomko et al., 2017*), suggest that the DNA translocase function of TFIIH is important to promote condensation. TFIIH is also important for recruiting factors that promote transcription, including the histone acetyltransferase KAT2A which controls histone H3K9 acetylation (*Sandoz et al., 2019*), a modification that is associated with increased nucleosome turnover (*Aygün et al., 2013*; *Zentner and Henikoff, 2013*). We speculate that the TFIIH complex regulates transient cycles of DNA opening and closing at Pol II-dependent promoters that do not result in transcription but help to maintain nucleosome-free regions required for condensin loading and function (*Gilchrist et al., 2010*; *Figure 5D*). This process may be part of or similar to mitotic transcriptional bookmarking observed in other cell types (*Palozola et al., 2019*). Importantly, we found that the condensation of both mouse and *Xenopus* sperm nuclei in *Xenopus* egg extracts require TFIIH activity (*Figure 5—figure supplement 1A*). This suggests a common fundamental mechanism that regulates condensin function. Interestingly, the TFIIH complex is present on chromosomes during mitosis in *Drosophila* embryos prior to the activation of transcription at the mid-blastula transition (*Cruz-Becerra et al., 2018*). Furthermore, in *Drosophila*, TFIIH interacts with the chromatin remodeling SRCAP complex (*Herrera-Cruz et al., 2012*), and mutation of the p52 TFIIH subunit causes chromosome condensation defects in mitotic neuroblasts (*Fregoso et al., 2007*). Thus, it is likely that the role of TFIIH in mitotic chromosome condensation during embryogenesis is

conserved. However, the chromatin landscape during embryogenesis is quite distinct from that of somatic cells (*Jukam et al., 2017*; *Ke et al., 2017*; *Du et al., 2017*), and therefore factors other than or in addition to TFIIH may regulate condensin function depending on cell type. In addition, the TFIIH complex has been reported to act independently of Pol II in various non-transcriptional processes (*Mizuki et al., 2007*; *Compe and Egly, 2016*), and therefore it is possible that TFIIH promotes a condensation-competent chromatin environment on its own or in association with unknown factors independent of transcription.

Seminal work from the Hirano lab demonstrated that mitotic chromatid structure can be reconstituted in vitro using *Xenopus* sperm and a limited set of purified factors that include core histones, histone chaperones (nucleoplasmin, Nap1 and FACT), condensin I, and topo II (*Shintomi et al., 2015*; *Shintomi and Hirano, 2021*). The lack of TFIIH dependence in this reconstituted system could be explained by either incomplete or highly dynamic chromatinization, which would expose DNA and eliminate the need for TFIIH (*Shintomi et al., 2015*; *Barrows and Long, 2019*). Interestingly, in vitro chromatid reconstitution in this system specifically requires the use of an egg-specific histone (*Shintomi et al., 2015*; *Shechter et al., 2009*), histone H2A.X-F (also known as H2A.X.3), and it was suggested that this variant is a better substrate of the FACT complex chaperone than is the canonical histone H2A, which is also present in egg extracts (*Shintomi et al., 2015*; *Wang et al., 2014*). Since the FACT complex promotes both nucleosome assembly and disassembly (*Zhou et al., 2020*), it may act in the reconstituted system to provide the nucleosome dynamics required for condensation that TFIIH provides in egg extracts. Since the FACT histone chaperone acts in concert with Pol II during transcription (*Petrenko et al., 2019*), and helps promote the loading of the condensin-related cohesin complex (*Garcia-Luis et al., 2019*), it will be important to investigate contributions of FACT to TFIIH-dependent condensation in egg extracts and in other systems. It is also possible that TFIIH regulates the access of other non-nucleosomal factors, not present in the reconstituted system, that either impede condensin loading or serve to restrict nucleosome mobility.

Although the regulation of nucleosome occupancy may be an important factor in condensin loading and function on chromatin, it is also possible that TFIIH promotes other changes necessary for condensation. Previous work has demonstrated that changes in DNA supercoiling and/or the creation of single-stranded DNA are important for condensin loading (*Sutani et al., 2015*; *Kim et al., 2021*). Indeed, TFIIH complex can act in all three processes: it helps to maintain nucleosome-free regions (*Gilchrist et al., 2010*), its DNA translocase activity exposes ssDNA during transcription initiation (*Aibara et al., 2021*; *Tomko et al., 2017*; *Fishburn et al., 2015*), and it promotes DNA supercoiling (*Aibara et al., 2021*; *Tomko et al., 2017*). TFIIH may also be involved in the removal of other non-nucleosomal barriers to condensin function, such as histone H1 or Pol II itself (*Choppakatla et al., 2021*; *Brandão et al., 2019*; *Rivosecchi et al., 2020*; *Pradhan et al., 2021*). Understanding how TFIIH alters the chromatin environment to promote condensin function and condensation should be the focus of future efforts.

## Maintenance of the condensed chromosome state is a dynamic process regulated by TFIIH

The mechanisms that maintain chromosome condensation are not well understood (*Paulson et al., 2021*). Our data show that continuous TFIIH activity is needed both to load condensin and to promote its function during the establishment and maintenance of chromosome condensation. In line with this, multiple studies have reported that the inactivation of condensin or its acute depletion from fully condensed chromosomes results in the rapid loss of condensation (*Hirano and Mitchison, 1994*; *Piskadlo et al., 2017*; *Samejima et al., 2018*; *Nakazawa et al., 2011*). Furthermore, continual ATP hydrolysis by condensin is required to maintain condensed chromosomes (*Kinoshita et al., 2015*; *Thadani et al., 2018*; *Elbatsh, 2019*). Thus, maintenance of mitotic chromosome structure is a dynamic and continuous process, and our data suggest that condensin access to DNA continues to be a critical step even after condensation is achieved. As the folding of chromatids is entropically unfavorable (*Goloborodko et al., 2016b*; *Vasquez et al., 2016*), it is worth noting that all the major factors required to promote condensation, including the XPB subunit of TFIIH, are ATPases (*Paulson et al., 2021*). Thus, our findings provide further support to the conclusion that mitotic chromosome assembly and maintenance are energy-dependent bidirectional processes opposed by nucleosomes and entropic forces (*Shintomi and Hirano, 2021*; *Hirano, 2014*).

# Materials and methods

## Key resources table

| Reagent type (species) or resource | Designation | Source or reference | Identifiers | Additional information |
|---|---|---|---|---|
| Antibody | Anti-XPB (rabbit polyclonal) | Novus Biologicals | Cat# NB100-61060, RRID:AB_925377 | IF (1:200), WB (1:1000) IP (0.24 µg antibody per µl beads and 0.24 µg antibody per µl extract) Immunodepletion (0.24 µg Antibody per µl beads and 1.44 µg Antibody per µl extract) |
| Antibody | Anti-XPB (mouse monoclonal) | Millipore | Cat# MABE1123 | IF (1:50) |
| Antibody | Anti-p62 (rabbit polyclonal) | Abcam | Cat# ab232982 | WB (1:400) |
| Antibody | Anti-XPF (rabbit polyclonal) | Gift of Puck Knipscheer, Hubrecht Institute *Klein Douwel et al., 2014* | Custom | WB (1:10,000) |
| Antibody | Anti-ERCC1 (rabbit polyclonal) | Gift of Puck Knipscheer, Hubrecht Institute *Klein Douwel et al., 2014* | Custom | WB (1:10,000) Immunodepletion (0.24 µg antibody per µl of beads and 1.44 µg of antibody per µl of extract) |
| Antibody | Anti-CAP-E Serum (rabbit polyclonal) | This study | HL7914 | Immunodepletion (24 µl of serum per µl extract and 3 µl of serum per µl of beads) IP (3 µl of serum per µl HSS and 3 µl of serum per µl of beads) Refer to 'Antibody Production' section in Methods |
| Antibody | Anti-CAP-D3 (rabbit polyclonal) | Gift of Alexei Arnaoutov and Mary Dasso, NICHD *Bernad et al., 2011* | Custom | IF (1:100) WB (1:400) |
| Antibody | Anti-CAP-C (rabbit polyclonal) | Gift of Alexei Arnaoutov and Mary Dasso, NICHD *Bernad et al., 2011* | Custom | IF (1:100) WB (1:500) |
| Antibody | Anti-CAP-E (rabbit polyclonal) | Gift of Alexei Arnaoutov and Mary Dasso, NICHD *Bernad et al., 2011* | Custom | IF (1:100) WB (1:1,000) |
| Antibody | Anti-CAP-G (rabbit polyclonal) | Gift of Susannah Rankin, Oklahoma Medical Research Foundation | OMRF191 | WB (1:5,000) |
| Antibody | Anti-Topo II (rabbit polyclonal) | Gift of Yoshi Azuma, University of Kansas *Pandey et al., 2020* | Custom | IF (1:200) |
| Antibody | Anti-Histone H3 (rabbit polyclonal) | Abcam | Cat# ab1791, RRID:AB_302613 | IF (1:400) WB (1:500) |
| Antibody | Anti-Histone (T3ph) (rabbit monoclonal) | Abcam | Cat# ab78351, RRID:AB_1566301 | WB (1:10,000) |
| Antibody | Anti-histone H4K12ac (mouse monoclonal) | Gift of Hiroshi Kimura, Tokyo Institute of Technology *Hayashi-Takanaka et al., 2015* | Custom | Immunodepletion (0.24 µg antibody per µl beads and 4.15 µg antibody per µl extract) |
| Antibody | Anti-α-Tubulin (mouse, monoclonal) | MilliporeSigma | Cat# T9026, RRID:AB_477593 | WB (1:20,000) |
| Antibody | Anti-CDK1T116Phos (rabbit polyclonal) | Cell Signaling Technology | Cat# 9,114 S, RRID:AB_2074652 | WB (1:500) |
| Antibody | Anti-SUPT16H (rabbit polyclonal) | Abcam | Cat# ab204343 | IF (1:100) |
| Antibody | Alexa Fluor 488 anti-rabbit IgG (donkey polyclonal) | Jackson ImmunoResearch | Cat# 711-545-152, RRID:AB_2313584 | IF (1:400) |
| Antibody | Alexa Fluor 647 anti-rabbit IgG (goat polyclonal) | Jackson ImmunoResearch | Cat# 111-605-144, RRID:AB_2338078 | IF (1:400) |
| Antibody | Alexa Fluor 488 anti-mouse IgG (goat polyclonal) | Jackson ImmunoResearch | Cat# 115-545-166, RRID:AB_2338852 | IF (1:400) |

*Continued on next page*

*Continued*

| Reagent type (species) or resource | Designation | Source or reference | Identifiers | Additional information |
|---|---|---|---|---|
| Antibody | IRDye 800 CW anti-rabbit IgG (H + L) (goat polyclonal) | LI-COR Biosciences | Cat# 926–32211, RRID:AB_621843 | WB (1:17,500) |
| Antibody | IRDye 680 LT anti-mouse IgG (H + L) (goat polyclonal) | LI-COR Biosciences | Cat# 926–68020, RRID:AB_10706161 | WB (1:17,500) |
| Antibody | IgG (rabbit polyclonal) | MilliporeSigma | Cat# I5006, RRID:AB_1163659 | Immunodepletion (used as control at 0.24 µg antibody per µl of beads) |
| Peptide, recombinant protein | xCAP-E C-terminal peptide | Vivitide | Custom | NH2-CSKTKERRNRMEVDK-COOH |
| Other | Protein A Dynabeads | Thermo Fisher Scientific | Cat# 10,002D | Immunodepletion and Immunoprecipitation |
| Chemical compound, drug | BS$^3$ (bis(sulfosuccinimidyl) suberate) | Thermo Fisher Scientific | Cat# A39266 | 4 mM |
| Chemical compound, drug | Triptolide | Tocris or MilliporeSigma | Tocris Cat# 3,253 MilliporeSigma Cat# T3652 | 50 µM |
| Chemical compound, drug | α-amanitin | Tocris | Cat# 4,025 | 54.4 µM |
| Chemical compound, drug | Actinomycin D | Tocris | Cat# 1,229 | 0.05 µM, 0.4 µM, 4.0 µM |
| Chemical compound, drug | THZ1 | Cayman | Cat# 9002215–1 | 30 µM |
| Chemical compound, drug | DRB | Cayman | Cat# 10010302 | 100 µM |
| Chemical compound, drug | RNase A | Thermo Fisher Scientific | Cat# EN0531 | 100 µg/ml |
| Chemical compound, drug | RNase H | Thermo Fisher Scientific | Cat# EN0201 | 0.2 Units/µl |
| Other | Hoechst Stain | Thermo Fisher Scientific | Cat# H3569 | 0.5 µg/ml |
| Other | Bio-Rad Trans-Blot Nitrocellulose | Bio-Rad | Cat# 170–4159 | Western blot |
| Biological sample (*Xenopus laevis*) | *Xenopus laevis*, 9 + cm mature females, pigmented | NASCO | Cat# LM00535M | Female, adult frogs |
| Biological sample (*Xenopus laevis*) | *Xenopus laevis*, 7.5–9 cm mature males, pigmented | NASCO | Cat# LM00715MX | Male, adult frogs |
| Software, algorithm | Fiji v2.1.0/1.53 f | Fiji | RRID: SCR_002285 | Image analysis and quantification |
| Software, algorithm | GraphPad Prism v9 | GraphPad | RRID: SCR_002798 | Graph creation and statistical analysis |
| Software, algorithm | NIS Elements AR 4.20.02 | Nikon Instruments | RRID: SCR_014329 | Capture and process chromatin images |
| Software, algorithm | Licor Image Studio v3.1 | LI-COR Biosciences | RRID: SCR_013715 | Capture and process Western blot images |
| Software, algorithm | Adobe Illustrator CC v25.4.3 | Adobe | RRID: SCR_010279 | Organize and prepare figures |

## Preparation of *Xenopus* egg extracts

Metaphase-arrested *Xenopus laevis* egg extract was prepared as previously described (*Haase et al., 2017*). This low-speed supernatant (LSS) extract was used to prepare mitotic high-speed supernatant (HSS) extract as previously described (*Maresca and Heald, 2006*). Briefly, LSS extracts were supplemented with energy mix (7.5 mM creatine phosphate, 1 mM ATP, 1 mM MgCl$_2$, 0.01 mM EGTA), 50 mM sucrose, and sea urchin cyclin Δ90 (*Groen et al., 2011*). Extracts were incubated at 21 °C for 15 min then spun for 2 hr at 200,000 *x g* at 4 °C. Post spin, the middle cytosolic layer was removed and spun once more for 30 min. The resulting supernatant was removed and aliquoted, and stored at –80 °C.

## Sperm nuclei preparation

*Xenopus* sperm nuclei were prepared as described previously (*Maresca and Heald, 2006*). Mouse (*Mus musculus*) sperm nuclei were prepared as described previously (*Shintomi et al., 2017*), with the

following modifications. Sperm were isolated from the cauda epididymis of ~6-month-old C57BL/6 J mice. To detach sperm tails from the nuclei-containing sperm heads, isolated sperm were resuspended in 2 ml of tissue-culture grade trypsin/EDTA (Thermo) and incubated for 5 min in a 37 °C water bath. Digestion was halted by resuspension of the mixture in 1 X PBS containing 10 X LPC (10 mg/mL each of leupeptin, pepstatin, and chymostatin). To isolate sperm heads from tails, the digestion mixture (resuspended in 1 X MH buffer [20 mM HEPES-KOH, pH 7.7, 2 mM MgCl$_2$] + 1.8 M sucrose) was layered onto a sucrose step gradient (containing 2.05 and 2.2 M sucrose, in 1 X MH buffer) and centrifuged for 45 min at 93,000 $x$ $g$ at 2 °C. After washing the interfaces with 1 X MH buffer, the pellet containing sperm heads was resuspended in 1 X MH +250 mM sucrose. Sperm nuclei were demembranated as described previously (*Shintomi et al., 2017*), and protamines were reduced in a solution of 1 X MH, 250 mM sucrose, 0.4% BSA, 1 X LPC, and 50 mM DTT for 2 hr in a room temperature water bath. Sperm were washed, resuspended, and stored as described.

## Chromatid assembly reactions and drug treatments

To monitor the assembly of chromatids, HSS egg extracts were first diluted with an equal volume of XBE5 buffer (10 mM HEPES-KOH [pH 7.7], 100 mM KCl, 5 mM EGTA, 5 mM MgCl$_2$, 50 mM sucrose) and supplemented with energy mix. Sperm nuclei isolated from *Xenopus* or mouse were then added to the diluted HSS extracts at a final concentration of 2000 sperm/μl, and incubated for the indicated times at 21 °C. Samples were taken at indicated timepoints for analysis via microscopy and western blot.

For inhibition experiments, chromatin assembly reactions were allowed to proceed for 25 min to allow for protamine exchange with maternal histones, and then drugs or an equal volume of DMSO were added and the extracts were further incubated at 21 °C for the indicated times. The final concentration of inhibitors was: 54.4 μM α-amanitin (Tocris), 50 μM triptolide (TPL; Tocris or Sigma), 30 μM THZ1 (Cayman), and 100 μM DRB (Cayman).

## RNase treatments

To determine the role of RNA in chromosome condensation, HSS extracts were either mock-treated (RNase-free water) or treated with RNase A (Thermo Fisher Scientific) at a final concentration of 100 μg/ml and incubated at 21 °C for 30 min prior to the addition of sperm. Sperm nuclei were then added to a final concentration of 2000 sperm/μl and the reaction was further incubated at 21 °C for up to 180 min in the presence of active RNase A. Samples were taken at indicated timepoints and processed for microscopy analysis. To analyze total RNA levels, RNA was isolated from 100 μl of HSS extracts according to the RNeasy kit 'RNA clean up' protocol (Qiagen), precipitated in the presence of glycogen (MilliporeSigma), and resuspended in 2 μl of RNase-free water. Samples were then diluted with RNA loading buffer (95% formamide, 10 mM EDTA pH 8, 0.025% SDS, bromophenol blue) and run on a 1% agarose gel in 0.5 X TBE, and visualized with SYBR safe DNA gel stain (Thermo Fisher Scientific). Note that total RNA levels in HSS extracts are low compared to LSS extracts due to the lower concentration of ribosomal RNAs (*Groen et al., 2011*).

To degrade RNA:DNA hybrids, HSS extracts were treated with RNase H (Thermo Fisher Scientific) or heat-inactivated RNase H (65 °C for 20 min) at a final concentration of 0.2 U/μl. Sperm nuclei were then added to a final concentration of 2000 sperm/μl and the reaction was incubated at 21 °C for up to 180 min. Samples were taken at indicated timepoints and processed for microscopy analysis.

## Antibody production

CAP-E antibodies were raised against a C-terminal peptide of *Xenopus laevis* CAP-E (CSKTKERRN-RMEVDK; synthesized by Vivitde) (*Hirano et al., 1997*). The peptide was then coupled to keyhole limpet hemocyanin protein and used to immunize rabbits (Labcorp).

## Immunodepletion of HSS extracts

Control rabbit IgG (MilliporeSigma, I5006), anti-XPB (Novus Biologicals, NB100-61060), anti-ERCC1 (a kind gift of Puck Knipscheer, Hubrecht Institute, *Klein Douwel et al., 2017*), or anti-histone H4K12ac (a kind gift of Hiroshi Kimura, Tokyo Institute of Technology, *Zierhut et al., 2014*) were bound and crosslinked to Protein-A Dynabeads as previously described (*Haase et al., 2017*). HSS extract was incubated with antibody-coupled beads at 4 °C for 30 min with occasional mixing, followed by two

additional rounds of depletion. For CAP-E depletion, CAP-E serum or pre-immune serum ("Mock") were bound to Protein-A Dynabeads without crosslinking, and HSS extract was incubated with antibody-coupled beads at 4 °C for 45 min, followed by one additional round of depletion. The volume ratio of beads to extract for each depletion were as follows: CAP-E serum (4:1), anti-XPB (2:1), anti-ERCC1 (2:1), and anti-H4K12ac (5.76:1). After depletion, the extract was used for chromatid assembly reactions as indicated.

## Immunoprecipitations

To provide samples for the analysis of XPB interaction partners by mass spec, XPB antibody (Novus Biologicals, NB100-61060) was incubated with Protein-A Dynabeads (Thermo Fisher Scientific, 10001D) and then crosslinked with BS³ (Thermo Fisher Scientific, A39266) at a ratio of 0.24 µg of antibody per µl of beads. A total of 300 µl of HSS extract was then incubated with an equal volume of anti-XPB beads for 60 minutes at 4 °C, with occasional mixing. Anti-XPB beads were then captured and washed four times with wash buffer (1 X PBS, 0.01% Tween-20), and boiled in 2 X sample buffer for 5 min at 95 °C.

For XPB and CAP-E immunoprecipitations, XPB antibody (Novus Biologicals, NB100-61060), control IgG antibody (MilliporeSigma, I5006), anti-CAP-E serum, or pre-immune ("Mock") serum were incubated with Protein-A Dynabeads at a ratio of 0.24 µg XPB or IgG antibody per µl of beads or 3 µl anti-CAP-E or pre-immune serum per µl of beads, and subsequently crosslinked with BS³. A total of 75 µl of HSS extract was then incubated with an equal volume of antibody-bound beads for 60 min at 4 °C, with occasional mixing. Antibody-bound beads were then captured and washed four times with wash buffer (1 X PBS, 0.01% Tween-20), and boiled in 2 X sample buffer for 10 min at 70 °C.

## Mass spectrometry

The immunoprecipitated proteins were separated by gel electrophoresis. Each lane was divided into 12 fractions and the proteins were in-gel digested with trypsin (Thermo Fisher Scientific) at 37 °C for 16 hr, as described (*Shevchenko et al., 2006*). Dried peptides were separated on a 75 µm x 15 cm, 2 µm Acclaim PepMap reverse phase column (Thermo) at 300 nL/min using an UltiMate 3000 RSLCnano HPLC (Thermo Fisher Scientific). Peptides were eluted into a Thermo Orbitrap Fusion mass spectrometer using a linear gradient from 96% mobile phase A (0.1% formic acid in water) to 55% mobile phase B (0.1% formic acid in acetonitrile) over 30 min. Parent full-scan mass spectra were collected in the Orbitrap mass analyzer set to acquire data at 120,000 FWHM resolution; ions were then isolated in the quadrupole mass filter, fragmented within the HCD cell (HCD normalized energy 32%, stepped ±3%), and the product ions were analyzed in the ion trap. Proteome Discoverer 2.4 (Thermo Fisher Scientific) was used to search the data against *Xenopus laevis* proteins from the PHROG database (*Wühr et al., 2014*; *Presler et al., 2017*) using SequestHT. The search was limited to tryptic peptides, with maximally two missed cleavages allowed. Cysteine carbamidomethylation was set as a fixed modification, with methionine oxidation and serine/threonine phosphorylation set as variable modifications. The precursor mass tolerance was 10 ppm, and the fragment mass tolerance was 0.6 Da. The Percolator node was used to score and rank peptide matches using a 1% false discovery rate.

## Western blots

Primary antibodies were diluted in Licor blocking solution with a final Tween-20 concentration of 0.1%. The following antibodies and antibody dilutions were used: anti-CDK1T161ph (9114S, 1:500, Cell Signaling Technology), anti-XPB (NB100-61060, 1:1000, Novus Biologicals), anti-p62 (ab232982, 1:400, Abcam), anti-α-Tubulin (DM1, 1:20,000, MilliporeSigma), anti-H3 (ab1791, 1:500, Abcam), anti-histone H3T3ph (ab78351, 1:10,000, Abcam), anti-XPF (a kind gift of Puck Knipscheer, Hubrecht Institute, *Klein Douwel et al., 2017*, 1:10,000), anti-ERCC1 (a kind gift of Puck Knipscheer, Hubrecht Institute, *Klein Douwel et al., 2017*, 1:10,000), anti-CAP-E (a kind gift of Alexei Arnaoutov and Mary Dasso, NICHD *Bernad et al., 2011*, 1:1000), anti-CAP-C (a kind gift of Alexei Arnaoutov and Mary Dasso, NICHD *Bernad et al., 2011*, 1:500), anti-CAP-D3 (a kind gift of Alexei Arnaoutov and Mary Dasso, NICHD *Bernad et al., 2011*, 1:400), and anti-CAP-G (a kind gift of Susannah Rankin, OMRF, 1:5000). Secondary antibodies (Licor) were used at a 1:17,500 dilution (goat anti-rabbit 800 nm and goat anti-mouse 680 nm), and membranes were scanned on the Odyssey imaging system (Licor).

## Microscopy and fluorescence quantification

To immunostain chromatin structures, extracts samples were fixed by diluting ten-fold with KMH buffer (20 mM Hepes-KOH pH 7.7, 100 mM KCl, 2.5 mM MgCl2) containing 4% formaldehyde and 0.1% Triton X-100, and incubated for 15 min at room temperature. To isolate chromatids/nuclei, the fixed reactions were layered onto cushions of KMH +30% glycerol in a 24-well plate, which contained poly-L-lysine–coated coverslips (No. 1) at the bottom of each well, and then spun at 4,614 x *g* for 12.5 min at 18 °C in a centrifuge (Eppendorf 5810 R). Cushion was removed and the coverslips were washed with TBS-TX twice for 5 minutes each, then blocked with Abdil (TBS +0.1% Tween20 +2% BSA + 0.1% sodium azide) for 30 minutes at 4 °C, then incubated in primary at 4 °C overnight unless otherwise noted. Chromatids/nuclei were then washed three times with AbDil and then incubated with secondary antibody for 2 hr. All washes and antibody dilutions were carried out with AbDil buffer. Chromatids/nuclei were stained with Hoechst 33258 prior to mounting in 80% glycerol +PBS medium. The following antibodies were used at the indicated dilutions: XPB (Millipore, MABE1123) 1:50, XPB (Novus Biologicals NB100-61060) 1:200, CAP-D3 (a kind gift of Alexei Arnaoutov and Mary Dasso, NICHD, *Bernad et al., 2011*) 1:100, CAP-C (*Bernad et al., 2011*) 1:100, CAP-E (*Bernad et al., 2011*) 1:100, Topo II (a kind gift of Yoshi Azuma, University of Kansas, *Pandey et al., 2020*) 1:200, SUPT16H (Abcam, ab204343) 1:100, histone H3 (Abcam, ab1791) 1:400. Alexa 488 (anti-rabbit, Jackson Immunoresearch), Alexa 488 (anti-mouse, Jackson Immunoresearch), and Alexa 647 (anti-rabbit, Jackson Immunoresearch) conjugated secondary antibodies were used at a 1:400 dilution for immunofluorescence detection.

All immunofluorescence was imaged with 0.2 µm step size using an Eclipse Ti (Nikon) comprised of a Nikon Plan Apo x100/1.45, oil immersion objective, a PlanApo x40/0.95 objective, and a Hamamatsu Orca-Flash 4.0 camera. Images were captured and processed using NIS Elements AR 4.20.02 software (Nikon), and analyzed in Fiji ImageJ. The acquired Z-sections of 0.2 µm each were converted to a maximum projection using NIS Elements and Fiji. Intensity of chromatin bound proteins was measured using Fiji by centering 9 × 9 and 13 × 13 pixel regions along multiple points of individualized chromatids and averaging the maximum intensity of each sample point to determine average intensity per chromatid. In cases where condensation was inhibited, chromatin masses were sampled to determine an average intensity. To correct for background fluorescence, the difference in intensities between the two regions was determined, and then made proportionate to the smaller region. This background value was then subtracted from the smaller region to determine chromatin bound protein intensity with background correction as previously reported (*Hoffman et al., 2001*).

## Condensation parameter

To quantify condensation of chromatin, 20 chromatin structures per condition were sampled with a 38 by 50 pixel region. Pixel intensities were recorded and binned. To measure the degree of condensation, the percentage of pixels below a threshold of 35% of the maximum image intensity were determined (*Maddox et al., 2006*). This percentage serves as the condensation parameter, and represents the changes in distribution of fluorescence intensity that result from condensation.

## Statistical analysis

All analyses were performed with a minimum number of 50 chromatids, or chromatin masses in cases where condensation was not achieved. This sample size was chosen to ensure a high ( > 90%) theoretical statistical power in order to generate reliable P values. All graphs and statistical analysis were prepared with GraphPad Prism. Fluorescence values from experimental conditions were compared to control conditions using an ordinary one-way ANOVA with Tukey's multiple comparison tests to determine significance. All graphs show the mean with error bars representing the S.D. unless otherwise indicated. At least two biological replicates (using two different preparations of egg extracts) were performed for each experiment and each confirmed similar behavior amongst the replicates.

## Acknowledgements

We thank Takashi Akera, Chongyi Chen, and Michael Lichten for critical reading of the manuscript, and members of the LBMB for comments and support, and A Arnaoutov, Y Azuma, M Dasso, H Kimura, P

Knipscheer, and A Nussenzweig for kindly providing reagents. We thank Jacob Paiano for assistance with mouse sperm purification, and Mia Rosenfeld for initial work with inhibitors.

## Additional information

### Funding

| Funder | Grant reference number | Author |
|---|---|---|
| National Cancer Institute | Intramural Research Program | Julian Haase<br>Richard Chen<br>Wesley M Parker<br>Mary Kate Bonner<br>Lisa M Jenkins<br>Alexander E Kelly |

The funders had no role in study design, data collection and interpretation, or the decision to submit the work for publication.

### Author contributions

Julian Haase, Conceptualization, Data curation, Formal analysis, Investigation, Methodology, Resources, Supervision, Validation, Visualization, Writing – review and editing; Richard Chen, Wesley M Parker, Conceptualization, Data curation, Formal analysis, Investigation, Methodology, Resources, Validation, Visualization, Writing – review and editing; Mary Kate Bonner, Data curation, Formal analysis, Methodology, Resources, Validation, Writing – review and editing; Lisa M Jenkins, Conceptualization, Data curation, Formal analysis, Funding acquisition, Investigation, Methodology, Project administration, Resources, Supervision, Validation, Visualization, Writing - original draft, Writing – review and editing; Alexander E Kelly, Conceptualization, Funding acquisition, Investigation, Methodology, Project administration, Resources, Supervision, Validation, Visualization, Writing - original draft, Writing – review and editing

### Author ORCIDs

Julian Haase http://orcid.org/0000-0001-5891-4241
Richard Chen http://orcid.org/0000-0002-5447-8646
Lisa M Jenkins http://orcid.org/0000-0003-1245-1338
Alexander E Kelly http://orcid.org/0000-0002-3395-6012

### Ethics

This study was performed in strict accordance with the care standards provided by the 8th edition of the Guide for the Care and Use of Laboratory Animals. African clawed frogs, Xenopus laevis, which were maintained and handled according to approved institutional animal care and use committee (NCI IACUC) protocol (LBMB-001-1) of the National Cancer Institute, which is an Association for Assessment and Accreditation of Laboratory Animal Care International (AAALAC) accredited research facility.

### Decision letter and Author response

Decision letter https://doi.org/10.7554/eLife.75475.sa1
Author response https://doi.org/10.7554/eLife.75475.sa2

## Additional files

### Supplementary files

• Transparent reporting form

### Data availability

The original files of the full raw unedited gels and blots and figures with the uncropped gels and blots with the relevant bands clearly labelled have been provided as Source Data files.

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
