## [Editor Report]

This paper reports the surprising observation that the general transcription factor TFIIH, but not transcription, is required for chromosome condensation in frog egg extracts. TFIIH may act by facilitating condensin localization and function. This opens up a lot of interesting new questions and lines of research that promise to add significantly to the field of chromosome biology. It will now be interesting to directly test the proposed mechanism of action, and to examine whether this role of TFIIH extends to somatic cells and other animals.

---

## [Decision Letter]

**Decision letter after peer review:**

Thank you for submitting your article "The TFIIH Complex is Required to Establish and Maintain Mitotic Chromosome Structure" for consideration by *eLife*. Your article has been reviewed by 3 peer reviewers, including Silke Hauf as Reviewing Editor and Reviewer #1, and the evaluation has been overseen by Jessica Tyler as the Senior Editor. The following individuals involved in review of your submission have agreed to reveal their identity: Rebecca Heald (Reviewer #2); Pim J Huis in 't Veld (Reviewer #3).

The reviewers have discussed their reviews with one another, and the Reviewing Editor has drafted this summary to help you prepare a revised submission.

All reviewers felt that you are reporting a novel, interesting, and likely impactful observation. The quality of the data was generally considered high, but several experiments require additional quantifications or controls.

Essential revisions:

1) Throughout the paper, images of "representative" chromosome morphologies are shown, but some form of quantification is required – either scoring condensation states subjectively, or – if possible – an objective (automated) quantification. Perhaps solidity measurements would help. It should also be stated how many times an experiment was replicated and how many structures were evaluated.

Quantifications would be helpful to strengthen the conclusions on some results that are currently difficult to interpret. For example, RNase H treatment and THZ1 treatment (Figure 1 – supplement 1) are interpreted as not showing an effect, but the chromatids look thinner in RNase H treatment, and somewhat uncondensed in the THZ1 treatment. In Figure 2A the +TPL at 25 min looks qualitatively different, as if it is taking longer for the effect to manifest. Is this the case?

2) It would be informative to see to what degree condensin and XPB co-localize on the chromosome axis at higher magnification, if this is possible.

3) All experiments, where no effect on condensation is observed (such as DRB and α-amanitin in Figure 1B or ERCC1 depletion in Figure 1G/H) would profit from showing that the drug or treatment did have an effect – to the extent that this is possible.

4) The authors could be yet more careful with the conclusions drawn. For example, the authors claim that TFIIH promotes both loading and activity of condensin from comparing the kinetics of chromosome morphology changes and condensin occupancy on chromatin. Other models would be equally consistent, e.g. TFIIH depletion could affect the binding or activity of some other molecule, which then causes condensin to fall off, like histone H1 which was shown to have an antagonistic relationship with condensin binding (see 10.7554/*eLife*.68918).

In the overall conclusions, the authors propose that TFIIH changes chromatin structure in a way that is conducive for condensin binding. This interpretation is similarly problematic since many proteins are competing for binding sites on chromatin and effects could be indirect and alternative models would not require changes in chromatin structure.

Additional suggestions (non-essential):

These are topics that came up in the discussion. If you can address any of them, that'd be great, but they are not required to meet the standard for publication.

1) Given the potential difficulties, this is not a requirement for publication, but the paper would obviously greatly profit from examining the role of TFIIH in condensation in other situations. For example, it would be interesting to know if the triptolide effect is specific to sperm chromosomes, or if mitotic chromosomes isolated from tissue culture cells, or formed from nuclei isolated from later developmental stages of embryogenesis, are similarly affected. Does triptolide block condensation in a somatic *Xenopus* cell line, or in a human cell line? Does TFIIH localize to mitotic chromosomes in somatic cells?

2) Given that the authors propose that TFIIH alters chromatin for condensin action, the paper would also be strengthened by examining nucleosome occupancy on sperm chromatin with and without TFIIH inhibition.

3) The interaction between condensin and TFIIH is potentially interesting. However, it seems that only a very small fraction of condensin is co-immunoprecipitated with XPB. Can the interaction be detected when immunoprecipitating condensin? Please also provide the full information from the mass spectrometry experiment. Was CAP-E the only condensin subunit detected? Do you have any evidence that the interaction is direct?

4) For the H3/H4 reduction + TPL, it would be very interesting to see not only the endpoint, but also the time course. Are chromosomes also resistant to decondensation in the first 5-10 minutes after TPL addition when histone levels are reduced?

5) It's striking that the TPL effects were abolished with mild (20-30%) H3/H4 depletions. Is it possible to determine the H3/H4 reduction that renders chromosome formation TPL insensitive in a more systematic way. What happens when H3/H4 are, for example, 5%, 10%, 20%, 40% or 80% reduced?

6) If extracts with histones are added to the histone-depleted chromosomes that condensed in the presence of TPL, would that result in decondensation? Does that depend on the inclusion of TPL in the add-back extract?

7) The authors could support the CDK7-independent conclusions of the paper by a) excluding abnormal CDK7 effects in the presence of TPL by inhibiting CDK7 simultaneously with XPB and b) excluding effects of TPL on overall CDK1 activity by immunoblotting for the phosphorylation of the activation loop of CDK1.

8) Are XPB levels reduced from chromatin when condensin I and/or condensin II is depleted from chromatin? Is it feasible to conduct this experiment?

9) The following paper might be interesting to discuss: https://www.nature.com/articles/s41467-019-09270-2

---

## [Author Response]

Essential revisions:1) Throughout the paper, images of "representative" chromosome morphologies are shown, but some form of quantification is required – either scoring condensation states subjectively, or – if possible – an objective (automated) quantification. Perhaps solidity measurements would help. It should also be stated how many times an experiment was replicated and how many structures were evaluated.

To quantify the chromosome morphologies in each experiment, we used the “condensation parameter” metric. This metric represents the percentage of pixels below a defined cutoff of fluorescence intensity for each image, and we feel it accurately reports the change in the shape of the fluorescence intensity distribution of chromatin that occurs during condensation. This approach was originally reported by the Desai and Oegema labs and used to quantify mitotic chromosome condensation in worm embryos and examine the effects of condensin depletion (Maddox et al., PNAS, 2006). We have provided this data both in the main figures, and in extensive scatter plots in: Figure 1, Figure 1—figure supplement 1, Figure 1—figure supplement 3, Figure 2, Figure 2—figure supplement 1, Figure 2—figure supplement 2, Figure 4, Figure 4—figure supplement 1, Figure 5, and Figure 5—figure supplement 2.

To put these metrics into perspective, we depleted condensins from extracts and determined the condensation parameters under this condition. As shown in Figure 1B and C, condensin depletion and TPL treatment have similar drastic effects on condensation (83% of pixels for DMSO at 180 min, 4% for condensin depleted, and 5% for TPL treated). We feel this further demonstrates the validity of the quantification method, and the similarities of the loss of condensation induced by triptolide treatment, TFIIH depletion, and condensin depletion.

We have also included information on the number of times experiments were replicated and how many structures were evaluated in the methods and figure legends of the manuscript.

Quantifications would be helpful to strengthen the conclusions on some results that are currently difficult to interpret. For example, RNase H treatment and THZ1 treatment (Figure 1 – supplement 1) are interpreted as not showing an effect, but the chromatids look thinner in RNase H treatment, and somewhat uncondensed in the THZ1 treatment. In Figure 2A the +TPL at 25 min looks qualitatively different, as if it is taking longer for the effect to manifest. Is this the case?

We believe that the condensation parameters now included in the manuscript should help address these concerns.

Regarding the RNase H treatment morphologies, we agree that the chromatids are a bit thinner than in wild type. However, this effect was also observed in the heat-inactivated RNase H condition as well (Figure 1—figure supplement 1C), and thus it is likely this is caused by one of the buffer components that the enzyme is provided in (e.g. KCl and/or EDTA), and not the activity of the enzyme.

We repeated the THZ1 treatment experiments multiple times and saw no obvious effect of the drug on chromosome morphology, and this is supported by the condensation parameter quantifications (Figure 1—figure supplement 3D). We have replaced the original images, which were from experiments in which the extracts were spotted directly onto a coverslip and fixed (“squashes”), with images from experiments in which we spun the chromosome-containing extracts through a glycerol cushion onto coverslips (Figure 1—figure supplement 1F), which provides a sharper view of individual chromosome morphology.

Regarding the TPL additions at 25 mins in Figure 2A, the new quantifications of the condensation parameters demonstrates that there is a drastic and rapid effect of TPL addition at all indicated timepoints, and that there is a slightly higher condensation parameter 5 mins after addition of TPL than at longer times (e.g. 7% at 30’ vs 4% at 185’ for the TPL @ 25’ condition; Figure 2—figure supplement 1). However, the controls demonstrate that the condensation parameter starts at around 40% at 25’, and plateaus around 80% at 180’. Thus, although there is a small and gradual decrease in condensation from 5 mins to 155 mins after TPL addition, the magnitude of the immediate effect of TPL addition is much larger.

2) It would be informative to see to what degree condensin and XPB co-localize on the chromosome axis at higher magnification, if this is possible.

We performed co-staining of XPB and CAP-E on chromatids, and this data is now included in Figure 3—figure supplement 1D.

3) All experiments, where no effect on condensation is observed (such as DRB and α-amanitin in Figure 1B or ERCC1 depletion in Figure 1G/H) would profit from showing that the drug or treatment did have an effect – to the extent that this is possible.

Data supporting the activity of the small molecules (THZ1, DRB, α-amanitin, and triptolide) is now provided in Figure 1—figure supplement 2. Although the first zygotic transcript is not produced until the 8-cell stage of *Xenopus* embryonic development (Jukam et al., 2017), multiple reports have demonstrated that active transcription occurs at the centromere repeats in egg extracts prior to zygotic activation (Grenfell et al., JCB,2016; Blower, Cell Reports, 2016; Perea-Resa et al., Mol. Cell, 2020). Indeed, we observed co-localization of both RNA Pol II S2ph (elongating) and RNA Pol II S5ph (paused) at centromeres. As expected, the elongation inhibitors α-amanitin and DRB blocked centromeric accumulation of S2ph, but not S5ph. Furthermore, THZ1 treatment, which inhibits the kinase responsible for S5ph that is upstream of the kinases required for S2ph, blocked centromeric accumulation of both forms of RNA Pol II. Likewise, triptolide, which inhibits the release of RNA Pol II from the pre-initiation complex, blocked centromeric accumulation of both forms of RNA Pol II. Thus, our new data show clear efficacy of each small molecule, and further support our claims of a specific role for the TFIIH complex in condensation.

We were unable to perform nucleotide excision repair (NER) assays to further evaluate the effects of ERCC1 depletion due to technical limitations. However, we have now quantified our ERCC1 depletion efficiency (Figure 1—figure supplement 2), and this level of depletion was sufficient to block NER in egg extracts using the same source of antibodies (Klein Douwel et al., EMBO, 2017; Mol. Cell, 2014).

4) The authors could be yet more careful with the conclusions drawn. For example, the authors claim that TFIIH promotes both loading and activity of condensin from comparing the kinetics of chromosome morphology changes and condensin occupancy on chromatin. Other models would be equally consistent, e.g. TFIIH depletion could affect the binding or activity of some other molecule, which then causes condensin to fall off, like histone H1 which was shown to have an antagonistic relationship with condensin binding (see 10.7554/eLife.68918).In the overall conclusions, the authors propose that TFIIH changes chromatin structure in a way that is conducive for condensin binding. This interpretation is similarly problematic since many proteins are competing for binding sites on chromatin and effects could be indirect and alternative models would not require changes in chromatin structure.

We agree with the reviewers and have extensively incorporated these suggestions throughout the revised manuscript. We discuss alternative models of TFIIH action and make clear that TFIIH could be acting directly or indirectly in these processes. Further, we have changed “chromatin structure” to “chromatin environment” to include both nucleosomes and chromatin binding proteins and enzymes.

Additional suggestions (non-essential):1) Given the potential difficulties, this is not a requirement for publication, but the paper would obviously greatly profit from examining the role of TFIIH in condensation in other situations. For example, it would be interesting to know if the triptolide effect is specific to sperm chromosomes, or if mitotic chromosomes isolated from tissue culture cells, or formed from nuclei isolated from later developmental stages of embryogenesis, are similarly affected. Does triptolide block condensation in a somatic *Xenopus* cell line, or in a human cell line? Does TFIIH localize to mitotic chromosomes in somatic cells?

Although we were unable to perform such experiments due to COVID and technical limitations, we agree that these are exciting experiments. Since submission of our manuscript, we discovered multiple earlier papers from the Zurita lab that reported defects in genome integrity, chromatin remodeling, and chromosome condensation in TFIIH mutants in *Drosophila* mitotic embryos and neuroblasts, and have incorporated these previous results in the Discussion. We feel this helps support a conserved role of the TFIIH in condensation during embryogenesis.

2) Given that the authors propose that TFIIH alters chromatin for condensin action, the paper would also be strengthened by examining nucleosome occupancy on sperm chromatin with and without TFIIH inhibition.

Unfortunately, we were unable to perform such experiments due to COVID and technical limitations, but hope to perform these exciting experiments in the future.

3) The interaction between condensin and TFIIH is potentially interesting. However, it seems that only a very small fraction of condensin is co-immunoprecipitated with XPB. Can the interaction be detected when immunoprecipitating condensin? Please also provide the full information from the mass spectrometry experiment. Was CAP-E the only condensin subunit detected? Do you have any evidence that the interaction is direct?

We performed an immunoprecipitation of condensin and we were unable to detect an interaction with XPB (Figure 3—figure supplement 1C). We have incorporated this data and our new data on XPB and condensin co-localization (Figure 3—figure supplement 1D) and the lack of an effect of condensin depletion on TFIIH chromatin levels (Figure 3—figure supplement 1A), in the Results section. We believe these findings demonstrate that any interaction of TFIIH and condensins is weak and not necessarily direct. This is in line with our H3/H4 partial depletion data (Figure 5) that shows that chromosomal condensin levels can be substantially restored in the absence of chromatin-bound TFIIH, and further supports our model that TFIIH acts upstream of condensin to maintain a chromatin environment that is conducive to condensin loading and function.

4) For the H3/H4 reduction + TPL, it would be very interesting to see not only the endpoint, but also the time course. Are chromosomes also resistant to decondensation in the first 5-10 minutes after TPL addition when histone levels are reduced?

We have included new data for this experiment detailing chromosome condensation 10 minutes after TPL or DMSO addition (Figure 5A and in Figure 5—figure supplement 2A). We found that chromosomes were resistant to decondensation in the H3/H4 reduction + TPL condition 10 minutes after drug addition, which we feel supports a primary effect of H3/H4 reduction in rescuing condensation.

5) It's striking that the TPL effects were abolished with mild (20-30%) H3/H4 depletions. Is it possible to determine the H3/H4 reduction that renders chromosome formation TPL insensitive in a more systematic way. What happens when H3/H4 are, for example, 5%, 10%, 20%, 40% or 80% reduced?

Unfortunately, we were unable to perform such experiments due to COVID and limited quantities of depletion-competent antibodies. We hope to perform these exciting experiments in the future.

6) If extracts with histones are added to the histone-depleted chromosomes that condensed in the presence of TPL, would that result in decondensation? Does that depend on the inclusion of TPL in the add-back extract?

Unfortunately, we were unable to perform such experiments due to COVID and limited quantities of depletion-competent antibodies. We hope to perform these exciting experiments in the future.

7) The authors could support the CDK7-independent conclusions of the paper by a) excluding abnormal CDK7 effects in the presence of TPL by inhibiting CDK7 simultaneously with XPB and b) excluding effects of TPL on overall CDK1 activity by immunoblotting for the phosphorylation of the activation loop of CDK1.

We have provided new data that exclude abnormal CDK7 effects in response to TPL treatment in Figure 1—figure supplement 1F. We have also provided new data that demonstrate no effect of TPL on CDK1 activation loop phosphorylation in extracts in Figure 1—figure supplement 1G.

8) Are XPB levels reduced from chromatin when condensin I and/or condensin II is depleted from chromatin? Is it feasible to conduct this experiment?

We have provided new data that demonstrate that depletion of condensins I and II have no effect on XPB levels on chromatin (Figure 3—figure supplement 1A).

9) The following paper might be interesting to discuss: https://www.nature.com/articles/s41467-019-09270-2

We have included this in the Discussion.